# Cross-sectional surveys of the amount of sugar, energy and caffeine in sugar-sweetened drinks marketed and consumed as energy drinks in the UK between 2015 and 2017: monitoring reformulation progress

Kawther M Hashem, Feng J He, Graham A MacGregor

Wolfson Institute of Preventive Medicine, Barts and The London School of Medicine and Dentistry, Queen Mary University of London, London, UK

**Correspondence to**
Kawther M Hashem;
k.hashem@qmul.ac.uk

## ABSTRACT

**Objectives** To investigate the sugar, energy and caffeine content of sugar-sweetened drinks marketed and consumed as energy drinks available in the UK.

**Study design** We carried out a cross-sectional survey in 2015 and 2017 of energy drinks available in the main UK retailers.

**Methods** The sugar (sugars g/100 mL), energy (kcal/100 mL), caffeine (mg/100 mL) and serving size were collected from product packaging and nutrition information panels of energy drinks available in the nine main UK grocery retailers, three health and beauty retailers and one convenience store.

**Results** The number of formulations (per 100 mL) and number of products (per serving) have fallen (from 75 to 49 and from 90 to 59) between 2015 and 2017, respectively. Energy drinks surveyed showed a 10% reduction in sugar, from 10.6 to 9.5 g/100 mL (P=0.011) and a 6% reduction in energy content (P=0.005) per 100 mL between 2015 and 2017. The average caffeine content of energy drinks, with a warning label, has remained high at 31.5±0.9 in 2015 and 31.3±1.0 mg/100 mL in 2017. Despite there being reductions, sugar, energy and caffeine content remain at concerning levels in 2017.

**Conclusions** To reduce the harmful impact of energy drinks, further reduction in sugar and a reduction in caffeine by reformulation are urgently needed. Other measures such as ban on the sale of energy drinks to children and smaller product sizes should also be explored, while warning labels should be kept. A reduction in sugar, energy and caffeine content and overall energy drinks consumption could be beneficial in reducing sugar, energy and caffeine intake of consumers of energy drinks.

## INTRODUCTION

In July 2015, the Scientific Advisory Committee on Nutrition (SACN) recommended that average free sugars (sugar) intake, across the UK population, should not exceed 5% of total energy intake.[1] This is in line with

### Strengths and limitations of this study

► This paper for the first time investigates and documents the sugar, energy and caffeine content of products marketed as energy drinks sold in the UK.
► The results demonstrate that the amount of sugar and energy has fallen, without technical issues, while caffeine levels remain high.
► This paper shows that serving sizes of energy drinks are large and should be reduced.
► The study was based on the levels provided on energy drinks packaging labels in store; hence, we relied on the accuracy of the data provided on the label.

the World Health Organisation's conditional new guideline on sugar intake.[2 3] SACN also advised that consumption of sugar-sweetened drinks, including energy drinks, should be minimised in children and adults,[1] because high intake of sugar is contributing to obesity, type 2 diabetes and dental caries,[3–11] major public health problems in the UK,[12–19] with significant healthcare costs.[20]

The average intakes in 2014 of sugar exceeded recommendations in all age groups.[21] The mean sugar intake in adults was 60 g per day and contributes to 12% of daily energy intake. In children the average sugar intake was 54 g (13%) per day in 4–10 year olds and 73 g (15%) per day in 11–18 year olds.[21]

Soft drinks are the main contributor of sugar intake in children (4–10 years) and teenagers (11–18 years) as well as the second main contributor in adults (18–64 years), contributing to 30%, 40% and 25% of sugar intake, respectively.[21] Within soft drinks, energy drinks are a booming subcategory.[22–25]

Products described as 'energy drinks' by the industry are typically glucose-based energy drinks: functional or stimulation energy drinks which claim a particular energy boost from caffeine, guarana, taurine, ginseng or other herbs or some combination of these ingredients, which are known to have stimulant properties and are distinct from sports drinks (which are often described as 'sports', 'isotonic' or 'hypotonic' and 'hypertonic').[22 26] Besides the health concerns around sugar content, the high levels of caffeine in some energy drinks are associated with chronic sleep loss, addiction/dependence, withdrawal and intoxication.[26–29]

In the UK, sales of energy drinks have increased by 155% between 2006 and 2014, from 235 to 600 million L, with an average per capita consumption of 9.4 L in 2014.[30] The leading brands' shares in the UK energy drinks market, by value in 2013–2014, were Red Bull (25%), own label (12%), Monster (10%), Relentless (6%), Rockstar (5%), Mountain Dew (2%), Boost (2%) and Emerge (2%).[31]

Energy drinks consumption in the UK is a growing problem, particularly among children and teenagers. In 2011, a study by the European Food Safety Authority found young people in the UK consumed more energy drinks than in other EU countries (3.1 L per person per month, compared with 2 L)[32] and market research data suggest a potential increase in purchase among children aged 10–14 years.[31]

The increase in consumption and popularity of energy drinks has raised concerns among the scientific community, governments and the general public about the health effects of these products, particularly among children and adolescents.[26 33 34] There have been two policies to tackle sugar and caffeine content of energy drinks. In terms of sugar content, a two-tiered Soft Drinks Industry Levy (SDIL) on the production and importation of sugar-sweetened drinks, including energy drinks, will be introduced in 2018.[35] The policy is intended to drive product reformulation and lower sugar consumption. Reformulation is commonly described as efforts by the manufacturer to lower the 'unhealthy' components (eg, saturated fat, trans fat, sugar, salt) of a product at the time of production, without worsening the profile of other ingredients (eg, increasing calorie content).[36] The reformulated products become a replacement of an existing product (eg, the same brand of energy drink with less sugar). The SDIL aims to incentivise manufacturers of sugar-sweetened drinks with more than 8 g of sugar, per 100 mL, to reduce the levels to below 8 g and pay a lower tax rate and manufacturers of drinks with more than 5 g of sugar per 100 mL and to lower the sugar levels to less than 5 g and pay no tax.[35]

In terms of caffeine content, the EU Food Information Regulation requires specific labelling for high caffeine drinks (over 150 mg/L).[37] The warning states that the product is not recommended for children. However, these products can be easily purchased and consumed by children.

This cross-sectional survey has been conducted to document levels of sugar and energy in energy drinks, as well as caffeine in the UK. The data available to monitor the energy drinks market, including whether reformulation is taking place, are generally the preserve of companies and not in the public domain. The main purpose of this research was to (1) report the variability in sugar and energy content in 2015 and 2017, (2) assess the sugar content in relation to the UK's new daily recommendation for sugar intake and by energy drinks brands, (3) quantify the levels of caffeine in energy drinks and determine the products on the market with the back of pack high-caffeine warning, (4) assess the number of products that will be taxed based on the criteria suggested in the SDIL[35] and (5) evaluate reformulation of energy drinks.

## METHODS

The data were collected from product packaging and nutrient information panels in January 2015 and January 2017. The survey was designed as a comprehensive survey of all energy drinks available in a snapshot in time, using one large outlet for each of the nine main grocery retailers, three health and beauty retailers and one convenience store. This study used the same study design and procedures as reported in our previous work on carbonated sugar-sweetened drinks, which excluded energy drinks.[38]

### Definition

Energy drinks were defined as any drink with 'energy', 'energise', 'energiser', 'caffeine' and 'stimulation' in the product name or description, for example, Red Bull Energy Drink, Monster Energy Drink, Relentless Origin Energy Drink and Tropicana Energy Mango and Guava with passion fruit or products with high-caffeine warning label including products not described as 'energy drinks' (such as Mountain Dew).

Since the focus of this study is on product reformulation, products labelled 'zero' or 'light' or 'no added sugar' were excluded. We also excluded products described solely as 'sports' drinks, which consist primarily of carbohydrates and electrolytes and are intended for athletes to rehydrate after exercise, for example, 'Lucozade Sports'.[22] However, 'Lucozade Energy' was included.

### Data collection

For each energy drink, the data collected included the company name, product name, pack weight, serving size, sugars (g), energy (kcal) and caffeine (mg) per 100 mL and per serving. Where data were not available per serving, it was calculated from pack size and per 100 mL data. Caffeine content was only collected in a subsample of the 2015 products.

Where products only labelled 'carbohydrate' content in 2015 but labelled 'sugars' content in 2017 and the content of all nutrients and ingredients were the same, it was assumed that the sugar content from 2015 was the same as 2017 (applied to only one product).

The data were double checked after entry and a further 5% of entries were checked against the original source in a random selection of products by the lead author.

## Stores

Data were collected from each of the major UK grocery retailers in store (Aldi, Asda, Lidl, Marks and Spencer, Morrisons, Sainsbury's, Tesco, The Co-operative and Waitrose), which altogether hold 93.2% of the grocery market share in the UK.[39] Three health and beauty retailers (Boots, Superdrug and Holland and Barrett) and one convenience store (Costcutter) were also included since they sell a wide variety of branded energy drinks.

## Analysis

### Per 100 mL

Some brands sell the same formulation in different serving sizes. Therefore, the 100 mL data only included an example of one formulation regardless of the different serving sizes.

### Per serving

The per-serving data included all the different serving sizes available apart from 1 L bottles. One-litre products were excluded from the per-serving analysis since it was deemed that the industry standardised serving of 250 mL was too little and that consumers are likely to overconsume. Any product with up to 500 mL can or bottle size was considered as one serving, regardless of what is stated on the packaging as a serving size for example, often a 500 mL bottle is split into two servings, but we consider that most consumers drink these drinks as one serving.

### Caffeine

Separate analysis was conducted on the products with the high-caffeine warning label (over 150 mg/L).

### High, medium and low criteria for sugars content

The sugar content was compared with the UK front of pack colour-coded labelling for drinks. Portion size criteria applied to portion/serving sizes greater than 150 mL. Colour coding is based on the following front of pack colour-coded nutrition labelling criteria (sugars: red/high >13.5 g/portion or >11.25 g/100 mL, amber/medium >2.5 to ≤11.25/100 mL, green/low ≤2.25 g/100 mL).[40]

In addition to the above analyses, the sugar content was compared with the UK's recommendation for sugar intake for adults (30 g/day) and children aged 7–10 years (24 g/day).[40] The highest energy-containing products were compared with women's daily energy intake (2000 kcal).[41] Also, we assessed the number of products that will be taxed based on the criteria suggested in the SDIL.[35]

## Statistical analysis

### Comparison among products within each survey

Independent samples Mann-Whitney U Test was used to compare the levels of sugar between supermarket own label and branded products.

### Comparison of the same products between the 2 years

For the purpose of assessing reformulation, only the products with data available in both surveys were included in this analysis. Paired t-test was used to examine whether there was a significant change in the sugar and energy content of energy drinks from 2015 to 2017.

Data are reported as mean, SD, range as indicated. Significance in all tests carried out was deemed significant as being $P<0.05$. All data were analysed using SPSS V.22.

## RESULTS

### Sugar, energy and caffeine content per 100 mL in 2015 and 2017

A total of 75 and 49 energy drinks met the per 100 mL inclusion criteria in 2015 and 2017, respectively. The average sugar content was 10.6±2.9 g and 9.7±3.0 g/100 mL, with a large variation in sugar content between different energy drinks ranging from 1.9 to 15.9 g and 2.1 to 16.0 g/100 mL in 2015 and 2017, respectively (table 1). There were no significant differences in sugar content between supermarket own label and branded products in 2015 (P=0.397) and 2017 (P=0.245). The product with the highest and lowest sugar content per 100 mL in 2015 and 2017 are highlighted in table 2.

The average energy content in energy drinks was 47±13 kcal/100 mL ranging from 10 to 70 kcal/100 mL in 2015 and 44±13 kcal/100 mL, ranging from 10 to 70 kcal/100 mL in 2017 (table 1). There were no significant differences in energy content between supermarket own label and branded products in 2015 (P=0.872) and 2017 (P=0.113).

Among all of the manufacturers with three or more formulations of energy drinks in 2015, Rockstar and Lucozade product ranges contained the highest average sugar and energy per 100 mL, respectively (table 3). However, in 2017 the Rockstar product range contained the highest average sugar and energy content per 100 mL (table 3), which suggests Lucozade have been reformulating their products.

Since not all 'energy drinks' have a high-caffeine content, we analysed the caffeine, sugar and energy content of the products with the warning label separately. A total of 23 and 39 energy drinks formulations had caffeine content labelled on pack in 2015 and 2017 (there may have been more that were labelled but not all were collected in 2015). The average caffeine content in energy drinks was 30.7±2.9 in 2015 and 31.6±0.8 mg/100 mL in 2017 (table 1). The levels of sugar, energy and caffeine were similar to the levels in the full sample of energy drinks.

### Sugar, energy and caffeine content per serving in 2015 and 2017

A total of 90 and 59 energy drinks products met the inclusion criteria in 2015 and 2017, respectively (table 4). The total number of products available has fallen between 2015 and 2017. The total number of products analysed

**Table 1** Sugar, energy and caffeine content in energy drinks per 100 mL

| Mean±SD (range) | 2015 | | | | 2017 | | | |
|---|---|---|---|---|---|---|---|---|
| | N (caffeine) | Sugar (g) | Energy (kcal) | Caffeine (mg) | N (caffeine) | Sugar (g) | Energy (kcal) | Caffeine (mg) |
| **All** | | | | | | | | |
| Own label | 13 (5) | 10.8±1.6 (9.6–15.9) | 48±6 (43–67) | 30.4±0.9 (30.0–32.0) | 9 (9) | 9.1±2.3 (4.9–12.8) | 39±8 (24–52) | 30.9±1.1 (30.0–32.0) |
| Branded | 62 (19) | 10.5±3.1 (1.9–15.6) | 47±14 (10–70) | 29.2±7.8 (0.0–32.0) | 40 (30) | 9.8±3.1 (2.1–16.0) | 45±13 (10–70) | 31.8±0.6 (30.0–32.0) |
| Total | 75 (24) | 10.6±2.9 (1.9–15.9) | 47±13 (10–70) | 29.4±6.9 (0.0–32.0) | 49 (39) | 9.7±3.0 (2.1–16.0) | 44±13 (10–70) | 31.6±0.8 (30.0–32.0) |
| **Labelled high caffeine** | | | | | | | | |
| Own label | 5 | 10.2±0.5 (9.6–10.9) | 45±2 (43–48) | 30.4±0.9 (30.0–32.0) | 9 | 9.1±2.3 (4.9–12.8) | 39±8 (24–52) | 30.9±1.1 (30.0–32.0) |
| Branded | 18 | 11.2±3.3 (3.0–15.6) | 49±12 (17–67) | 30.8±3.3 (18.0–32.0) | 30 | 10.5±2.9 (2.1–16.0) | 45±12 (10–67) | 31.8±0.6 (30.0–32.0) |
| Total | 23 | 11.0±2.9 (3.0–15.6) | 48±11 (17–67) | 30.7±2.9 (18.0–32.0) | 39 | 10.2±2.8 (2.1–16.0) | 43±11 (10–67) | 31.6±0.8 (30.0–32.0) |

per serving is different to the per 100 mL data because some brands sell the same formulation in different serving sizes. The serving data included all the different serving sizes available apart from 1 L bottles. The serving size varied from 150 to 500 mL in 2015 and 250 to 500 mL in 2017. Most products were in a 500 mL can/bottle (42% in 2015 and 47% in 2017).

The average sugar content in energy drinks was high 41.1±17.9 in 2015 and 38.5±18.2 g/serving in 2017 (table 4), more than the entire maximum daily recommendation for sugar intake in the UK for an adult (30 g). Indeed, 59% of products in 2015 and 54% in 2017 exceeded the maximum UK's recommendation for sugar intake per serving for an adult (30 g/day). Additionally, 86% in 2015 and 78% in 2017 of products exceeded the maximum daily recommendation for sugar intake for a child aged 7–10 years (24 g/day). About 96% and 95% would receive a 'red' (high) label for sugars per serving (>13.5 g/serving) in 2015 and 2017, respectively.

The serving size was significantly larger in branded versus supermarket own label products in 2015 (P=0.003) and 2017 (P=0.003), and as a result, the branded products contained on average higher levels of sugar compared with supermarket products per serving in 2015 (P=0.028) and 2017 (P=0.026) (figure 1 and table 4).

The average energy content in energy drinks was 185±81 kcal and 176±76/serving in 2015 and 2017, respectively. There was a large variation in energy content between different energy drinks ranging from 50 to 350 and 50 to 335 kcal/serving in 2015 and 2017, respectively (figure 2 and table 4).

The energy drink products with the highest energy content in 2015 and 2017 can contribute up to 17%–17.5% of a woman's daily energy intake (2000 kcal).[41]

A total of 22 and 43 energy drinks formulations had high-caffeine labels on pack (table 4) in 2015 and 2017, respectively.

### Soft drink industry levy in 2017
A total of 36 products (73%) would be taxed at the higher SDIL rate (>8 g/100 mL), nine products (18%) at the lower rate (8–5 g/100 mL) and four products (8%) would not be taxed (<5 g/100 mL).[35]

### Comparison of the same products between the 2 years
The overall number of energy drinks formulations has decreased between 2015 and 2017, from 75 to 49 products.

### Reformulation
There were 30 products surveyed repeatedly in 2015 and 2017. The average sugar content per 100 mL for these products was 10.6±3.2 g in 2015 and 9.5±3.3 g in 2017 (P=0.011 for comparison between the 2 years). This represents a reduction of 10%. The sugar content has fallen in 12 products, remained the same in 16 and has increased in 2. The average energy content per 100 mL was 47±14 kcal in 2015 and 44±15 kcal in 2017, (P=0.005

**Table 2** Highest and lowest sugar energy drink per 100 mL in 2015 and 2017

| | 2015 | | 2017 | |
|---|---|---|---|---|
| | **Product name** | **Sugar (g)** | **Product name** | **Sugar (g)** |
| Highest | Sainsbury's Orange Energy Drink | 15.9 | AG Barr—Rockstar Punched Energy+Guava Tropical Guava Flavour | 16.0 |
| Lowest | Monster Rehab Green Tea Energy | 1.9 | Monster Rehab Tea+Still Lemonade+Energy | 2.1 |

for comparison between the 2 years) representing a 6% reduction.

There were 12 products surveyed repeatedly in 2015 (31.5±0.9) and 2017 (31.3±1.0 mg/100 mL) displaying the high caffeine warning label. The caffeine content of these products stayed the same, apart from one product, where the caffeine content was reduced, from 32 mg to 30 mg.

Note that these averages are slightly different from those when all products were included in each year, and this trend analysis reflects reductions made in the same products rather than the overall products available, giving a better reflection of reformulation, for a full list of products refer to online supplementary file.

## DISCUSSION

These cross-sectional surveys have documented the levels of sugar, energy and caffeine in energy drinks in the UK between 2015 and 2017, a topic which has not been previously documented for easy access by researchers and policy-makers. It will allow for evaluation of trends in the energy drinks market in the future.

The surveys have shown early changes in the energy drinks market. Formulations (per 100 mL) and number of products (per serving) have fallen between 2015 and 2017. This may be due to the pressure on the soft drinks industry to reduce sugary drinks offerings.[42] Furthermore, the energy drinks surveyed showed a small but statistically significant reduction in sugar content; this is likely due to the 12 products which have reduced sugar content between 2015 and 2017, probably as a result of reformulation due to the impending SDIL.[35] Manufacturers of the reduced products have either only reduced sugar or have alternatively reduced sugar and replaced it with non-caloric sweeteners without changing the product name, for example, by calling the product 'light' and so on. Given the volume consumed,[22] even small reductions in sugar content of energy drinks would have a significant impact on sugar and energy intake among regular consumers of energy drinks.[43]

**Table 3** Sugar and energy content in energy drinks by manufacturer per 100 mL

| | 2015 | | | 2017 | |
|---|---|---|---|---|---|
| **Manufacturer** | **N** | **Sugar (g) mean±SD (range)** | **Manufacturer** | **N** | **Sugar (g) mean±SD (range)** |
| Rockstar | 8 | 14.1±1.1 (12.1–15.6) | Rockstar | 5 | 14.2±1.5 (12.0–16.0) |
| Lucozade | 10 | 12.5±1.5 (8.7–14.0) | Relentless | 5 | 11.1±2.4 (7.5–14.0) |
| Sainsbury's | 4 | 12.1±2.6 (10.6–15.9) | Red Bull | 3 | 11.0±0.0 (11.0–11.0) |
| Relentless | 5 | 10.6±0.6 (10.1–11.6) | Lidl | 5 | 10.2±1.7 (8.8–12.8) |
| Asda | 4 | 10.3±0.5 (9.6–10.7) | Lucozade | 6 | 8.6±3.4 (4.3–13.0) |
| KX | 3 | 10.2±0.2 (10.1–10.5) | Monster | 8 | 8.5±2.8 (2.1–11.0) |
| Little Miracles | 3 | 6.4±0.2 (6.2–6.6) | Little Miracles | 3 | 5.7±0.5 (5.1–6.1) |
| Monster | 7 | 6.4±4.2 (1.9–11.0) | | | |
| **Manufacturer** | **N** | **Energy (kcal) mean±SD (range)** | **Manufacturer** | **N** | **Energy (kcal) mean±SD (range)** |
| Lucozade | 10 | 63±5 (57–70) | Rockstar | 5 | 60±7 (51–67) |
| Rockstar | 8 | 60±5 (52–67) | Lucozade | 6 | 52±15 (35–70) |
| Sainsbury's | 4 | 53±9 (48–67) | Red Bull | 3 | 45±1 (45–46) |
| Asda | 4 | 46±2 (43–48) | Relentless | 5 | 46±10 (31–58) |
| Relentless | 5 | 44±2 (43–48) | Lidl | 5 | 42±7 (36–52) |
| KX | 3 | 44±2 (42–46) | Monster | 8 | 37±12 (10–48) |
| Monster | 7 | 29±18 (10–48) | Little Miracles | 3 | 25±2 (23–26) |
| Little Miracles | 3 | 27±1 (26–27) | | | |

**Table 4** Sugar, energy and caffeine content in energy drinks per serving

| Mean±SD (range) | 2015 | | | | 2017 | | | |
| --- | --- | --- | --- | --- | --- | --- | --- | --- |
| | N (caffeine) | Sugar (g) | Energy (kcal) | Caffeine (mg) | N (caffeine) | Sugar (g) | Energy (kcal) | Caffeine (mg) |
| **All** | | | | | | | | |
| Own label | 12 (4) | 30.8±11.1 (23.9–54.5) | 137±49 (107–245) | 76.2±2.5 (75.0–80.0) | 8 (8) | 27.3±15.6 (12.3–64.0) | 116±61 (60–260) | 87.5±29.4 (75.0–160.0) |
| Branded | 78 (19) | 42.7±18.3 (9.5–78.0) | 193±83 (50–350) | 115.7±47.1 (0.0–160.0) | 51 (35) | 40.2±18.0 (9.9–80.0) | 185±75 (50–335) | 132.7±36.6 (75.0–160.0) |
| Total | 90 (23) | 41.1±17.9 (9.5–78.0) | 185±81 (50–350) | 108.9±45.3 (0.0–160.0) | 59 (43) | 38.5±18.2 (9.9–80.0) | 176±76 (50–335) | 124.3±39.3 (75.0–160.0) |
| **Labelled high caffeine** | | | | | | | | |
| Own label | 4 | 25.5±1.6 (23.9–27.3) | 113±6 (107–120) | 76.3±2.5 (75.0–80.0) | 8 | 27.3±15.6 (12.3–64.0) | 116±61 (60–260) | 87.5±29.4 (75.0–160.0) |
| Branded | 18 | 46.1±22.4 (9.9–78.0) | 199±89 (56–335) | 122.1±39.0 (75.0–160.0) | 35 | 44.2±18.1 (9.9–80.0) | 189±75 (50–335) | 132.7±36.6 (75.0–160.0) |
| Total | 22 | 42.4±21.7 (9.9–78.0) | 183±87 (56–335) | 113.8±39.5 (75.0–160.0) | 43 | 41.1±18.7 (9.9–80.0) | 175±77 (50–335) | 124.3±39.3 (75.0–160.0) |

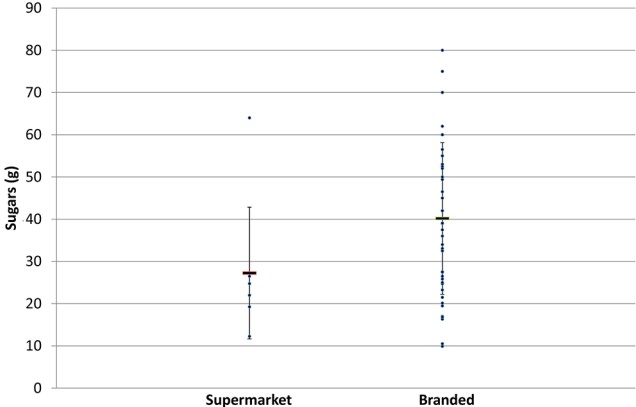

**Figure 1** Sugar content in supermarket and branded energy drinks (g/serving) in 2017.

This survey suggests the early success of the impending SDIL. Even though the SDIL is yet to come in, manufacturers have started to reformulate their products in advance to avoid paying the levy. However, we are yet to see if the SDIL will have an impact on overall consumption of energy drinks by reducing sales.[44] Other countries have imposed taxes on soft drinks, including energy drinks, and have seen reductions in sales.[45 46]

Despite this, sugar content remains at concerning levels, 95% would receive a 'red' (high) label for sugars per serving in 2017. There was also a large variation in sugar content between different energy drinks ranging from 2.1 to 16.0 g/100 mL, which suggests further reductions are possible.

As well as the concerns around the sugar and energy content of energy drinks, there are also concerns about caffeine levels, particularly in the products labelled as high caffeine. In 2017, average caffeine content in energy drinks was 124.3±39.3 mg/serving among the 43 products labelled, almost equivalent to two cups of coffee or four cans of cola.[37] There is also some evidence to suggest that the caffeine content increases sugar-sweetened drinks consumption further and therefore sugar intake too.[47] Since children and teenagers are the main consumers of energy drinks,

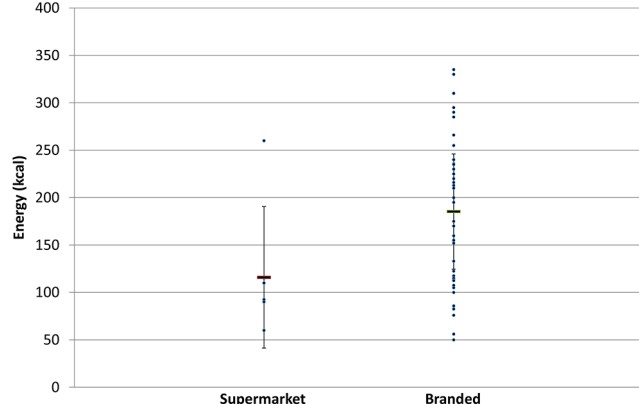

**Figure 2** Energy content in supermarket and branded energy drinks (g/serving) in 2017.

manufacturers should consider reducing levels of caffeine—again through reformulation. The removal of caffeine also allows for the removal of 10.3% of sugar without affecting taste, which has the potential to reduce body weight of adults by 0.6 kg, without any change in sugar-sweetened drinks consumption.[48]

It important to note that not all manufacturers label the amount of caffeine on their energy drinks, and this is likely because their products do not exceed the cut-off for adding the warning label.[37] Among the manufacturers that do, a warning is stated on the pack, acknowledging the potential danger these products can pose to children. However, these products can be easily purchased and consumed by children. Perhaps the UK should consider restrictions like other countries. In Lithuania, sales of energy drinks (containing at least 150 mg of caffeine per litre) are banned to under 18 years of age.[49] A similar law applies in Sweden where sales of energy drinks are banned to children under 15 years of age and sales of some energy drinks are restricted to pharmacies.[26]

Furthermore, serving sizes of energy drinks matter since the larger the serving size, the more sugar, energy and caffeine is consumed. Indeed, typical serving sizes of energy drinks are larger than other sugar-sweetened drinks[38]; this survey showed that the most common serving size was 500 mL. The average sugar content in energy drink in both surveys per serving was more than an adult's entire maximum daily recommendation for sugar intake in the UK. Likewise, 86% in 2015 and 78% in 2017 of the products exceeded the maximum daily recommendation for sugar intake for a child aged 7–10 years (24 g/day).[1] The products with the highest sugar and energy content per serving could contribute up 17.5% of a woman's total daily energy intake. Therefore, to reduce the amount of sugar, energy and caffeine consumed from energy drinks, there is scope for introducing restrictions on larger serving sizes (500 mL bottles and cans). This is a growing problem, which must be tackled, particularly since the Grocer recently reported an increase in sales of larger cans of energy drinks on the UK market.[23]

Our study was based on sugar, energy and caffeine content data provided on the available energy drinks packaging labels in store on the dates of collection; hence, we relied on the accuracy of the data provided on the label and the availability of products in store. It is assumed that the manufacturers provide accurate and up-to-date information in line with EU regulations. However, future studies should include sugar and energy content determined through laboratory analysis to achieve a better understanding of the true sugar and energy content and breakdown of the types of sugars. Furthermore, an analysis of what are the best selling products in the UK would allow for an accurate estimate of the potential impact of reformulation.

Also, the data collection process of caffeine content was slightly different in the two surveys. In 2015, caffeine was collected in a subsample of products, since the aim was not to collect caffeine at the time, but the data that were collected were included in the analysis, this may not give a full reflection of caffeine content of all products on the market in 2015.

Still, the results of this study are relevant and serve to document the sugar, energy and caffeine content of energy drinks sold the in UK, underpinning future studies and providing evidence for the treasury, policy-makers and the soft drinks industry, who are reformulating.[50 51]

## CONCLUSION

Factors such as an increase in sales,[24 25 30] concerning levels of sugar, energy and caffeine (assessed in this study), ease of access,[33] large serving sizes, as well as limited regulation have created an environment where energy drinks could pose a major threat to public health.

To reduce the impact of energy drinks, sugar (and therefore energy) reformulation should continue and begin regarding caffeine content. Other measures such as a ban on the sale of energy drinks to children should be explored, while warning labels should be kept. Can and bottle sizes of energy drinks should also be decreased to reduce the overall consumption of sugar, energy and caffeine. A reduction in sugar, energy and caffeine content and overall energy drinks consumption could be beneficial in reducing sugar, energy and caffeine intake of consumers of energy drinks.

**Contributors** KH conducted the research; KH and FJH analysed the data; KH wrote the first draft of the manuscript and all authors contributed to the interpretation of the results and revision of the manuscript and approved the final version. GAM had primary responsibility for the final content.

**Competing interests** None declared.

**Patient consent** Obtained.

**Provenance and peer review** Not commissioned; externally peer reviewed.

**Data sharing statement** No additional data are available.

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
