## [Reviewer comments · BMJ Open]

ARTICLE DETAILS

TITLE (PROVISIONAL)	Cross-sectional surveys of the amount of sugar, energy and caffeine in sugary soft drinks marketed and consumed as energy drinks in the UK between 2015 and 2017 - Monitoring reformulation progress
AUTHORS	Hashem, Kawther; He, Feng; MacGregor, Graham

VERSION 1 – REVIEW

REVIEWER	Laura Keaver Institute of Technology Sligo, Ireland
REVIEW RETURNED	13-Jun-2017

GENERAL COMMENTS	Thank you for the opportunity to review the paper “Cross-sectional surveys of the amount of sugars, calories and caffeine in energy drinks in the UK between 2015 and 2017” and I apologize to the authors and editors for any miscomprehension of the manuscript content, if any. This study seems to have been conducted with a particular objective in mind: to use data on the sugar content of energy drinks in UK grocery stores as a means to argue for product reformulation policies. The title therefore is misleading as to the content of the paper as calories and caffeine are not adequately discussed. The only reason for a survey so quickly (2017) after the previous one (2015) was to check for reformulation so maybe a paper with a clear focus on reformulation could be developed. Also the fact that there are 31 less drinks available after only 2 years is not referred to at all. An even more pressing point and the reason why I chose to reject the article is that the term “energy drinks” has been used inappropriately – while there are many definitions as to exactly an energy drink encompasses, one similarity in all definitions is that they contain caffeine – something which the authors themselves state in line one of their introduction however in the methodology it is stated that any sugar sweetened drink which used the term energy, energise, stimulate or similar on its packaging was included. This is essentially then an additional review of a subset of soft drinks but not actually energy drinks. Any additional work on this needs to consider what to define this group as. Some additional concerns, in order of the layout of the manuscript: 1. Abstract – It is not clear what the numbers in the bracket refer to in the study design, I assume it is products surveyed and therefore this would be more appropriate in results. Reference to the maximum daily recommendation for sugar is made in the results but then adults are used in the conclusions – be consistent. In addition, as with the closing conclusion of the paper it
---

is a very broad and unsupported claim that “reducing sugar content of the drinks would be large in reducing both sugar and calorie consumption and therefore a reduction in obesity, type 2 diabetes and dental caries”

2. There is a lack of referencing throughout the paper, I have included the lines here: (page 3 lines 23, 24; page 4 lines 15, 18, 35; page 5 line 12; page 6 line 3, page 7 line 16; page 8 line 44, page 9 line 34; page 12 line 5, 16, 19, 56; page 13 line 17, 25, 31, 44, 52, 54)

3. Introduction – Focusses solely on sugar content therefore not sure why calories and caffeine are included in title – seems like an afterthought. Using the term free sugars may be confusing to a lay audience, after defining free sugars I think it would be okay to state that you will refer to these as sugar throughout. There is mix of tenses in some sentences e.g. lines 20-21.

4. Method – the definition of energy drinks is not consistent with that of your first line in the introduction or of universally understood definitions. For serving size even when stated on a 500ml bottle label that 250ml was the serving size it was decided to include 500ml as a serving size however when dealing with the 1 litre bottles a 250ml serving size were used, aren't people likely to also overconsume with this size bottle?

5. Results – Including the same brand in different serving sizes will not give an accurate picture of reformulation as one brand with 4 different serving sizes will give the appearance of several drinks changing their content. I think in future using per 100ml while stating the serving sizes available may be more comprehensive and easier to follow. There are five pages of results which is far too much and it is very repetitive and seems unclear in what it really wants to show. You have also included some material which may be more suited to the discussion in this section. On page 7 line 43 sugar should read calorie.

When comparing the average sugar content to the maximum daily recommendation for sugar intake (page 10 line 49 onwards), you mention among the 65 products – it is not clear which 65 you are referring to as you have presented so many products. Also you need to include the word sugar in the brackets for line 52.

Including a list of the products as an appendix may have been useful

6. Discussion – this is started as energy drinks consistently surveyed, I'm not sure that twice constitutes consistently. You also state a significant reduction in sugar content however, my understand is this really refers to one brand with several different serving sizes?

In line 18 page 14 it is mentioned that even small reductions in sugar content of energy drinks would have a significant impact on sugar and calorie intake of the population but this is not substantiated with evidence.

7. Conclusion – There is information here not previously mentioned anywhere in the paper such as pervasive and misleading marketing targeted at young people.

“This study indicates there has been a reduction in sugar content of energy drinks in the UK” – there has only been a reduction in some or most likely one main brand. In addition, regarding the last line of

	the conclusion it is very naïve statement to assume that product reformulation would lead to a very large reduction in sugar and calorie consumption and therefore a reduction in obesity, type 2 diabetes and dental caries. Obesity in particular is a multi-faceted and complex condition and it will take much more than reformulation of energy drinks to make an impact on levels. Also it is not clear that this would change population dietary patterns as substitution is likely. I think that the authors may be able to reuse some of the data to look solely at reformulation and to be more clear about that at the beginning, however in its current format I don't think this paper is suitable for publication.
--	--

REVIEWER	Shelina Visram Associate professor in public policy and health Durham University, UK
REVIEW RETURNED	26-Jun-2017

GENERAL COMMENTS	Thank you for the opportunity to review this manuscript, which I found interesting and relevant to my own work. There has been little research to date on the contents of commercial energy drinks, which continue to increase in popularity particularly amongst young consumers. The authors' analyses appear to be robust and the results are reported in some detail. However, there are a number of areas where I believe the manuscript could be strengthened in order to make it suitable for publication. INTRODUCTION:  • The text in this section could be reordered to tell a clearer story, beginning with the SACN recommendation, a clear definition of free sugars, the role of soft drinks and then drilling down to energy drinks as a specific sub-category of soft drink. • The included text on energy drinks has largely been taken from either a previous BMJ Open paper (Visram et al 2016) or an FRC briefing paper (Visram and Hashem 2016) and therefore needs to be replaced, reworded or cited as direct quotations. • The authors refer to 'Another study by EFSA' (p. 3, line 40) but this is actually the same study as that cited in the previous sentence. The paper by the Nomisa-Areté Consortium is the original source for this study. • The terms 'free sugars' and 'sugars' are used throughout the manuscript. While this may be technically correct, 'sugar' (singular) is more appropriate. For example, it should be 'sugar-sweetened beverages', not 'sugars-sweetened beverages'. • This section makes a convincing case for investigating the sugar and calorie content of energy drinks, in light of the SACN recommendation and planned sugar levy. However, no rationale is given for examining data in relation to caffeine. METHODS:  • The data collection process is unclear – how exactly were data collected, e.g. in store or online? Using a predefined template? This section states that 'All data was double checked after entry' (p.5, line 29) – entry into what? Checked by whom? • What was the rationale for including 'beauty and health food
---

retailers' in this study? I fear this may have skewed some of your results (see below).

- 'Convenience store' needs to be added in brackets after 'a Costcutter store' (p.5, line 42).
- Has the term 'supermarket own brand' been used to refer to 'own brand' products on sale in convenience and health food stores as well as supermarkets? If so, this is confusing and needs rewording. If not, what was the rationale for only including supermarket products?
- Related to a previous point, it should be stated explicitly somewhere in the manuscript that sugar was the primary concern in this study, with caffeine a secondary concern. This would then help to clarify the rationale for excluding 'no added sugar' drinks, which are still likely to be high in caffeine and in need of reformulation.
- On reading your definition of an energy drink, readers may be confused if they are familiar with products such as Red Bull or Monster which do not seem to have 'energy' in the name. Please include one or two examples to reduce confusion, e.g. 'Monster energy drink'.
- You state that you excluded isotonic or sports drinks but presumably this was not true if they met the inclusion criteria, e.g. did you include Lucozade Energy and, if not, why not?
- The sub-sections on 'Serving size' and 'Per 100ml verses per serving' could be combined (also 'verses' should be changed to 'versus').
- The tense used in the sub-section on 'High, medium and low criteria' should be changed from present/future to past tense.

RESULTS:

- Throughout this section (and the manuscript as a whole), the information on caffeine reads as an afterthought. It would be preferable to either leave it out altogether or do more to highlight its importance. There is no mention of caffeine in the tables or figures.
 - It is not clear why the numbers of products surveyed are different in the sub-sections on 'per 100ml' and 'per serving'. Did some of the product labels lack information on sugar and calories per 100ml? If so, it would be useful to identify the types of products affected (given that they should all presumably be subject to the same food labelling regulations).
 - Similarly, it would be helpful to know more about products that were not labelled with their caffeine content or the 'high caffeine' warning. Energy drink labels are required to include this information by EU legislation in place since 2011.
 - The low values given here seem very low and I wonder if these reflect the inclusion of products sold in health food stores, which seem like a different category of products that are less likely to be used by children and adolescents. Were they sold primarily as food or exercise supplements?
 - It would be good to know more about the products with very low and very high levels of sugar, as well as those that were reformulated between the two time periods.
- The tables only provide averages for own brand and branded products, which are useful, but it would be good if they could also include information on the top and bottom point of each range.
- This section is a little hard to read and I wonder if it would be easier to follow if sub-sections were combined across the two years, i.e. sugar / calories / caffeine / per 100ml in 2015 vs 2017 and per serving size in 2015 vs 2017. There is a separate sub-section comparing the same products between the two years but it is also interesting to note overall changes across all products in this period.

	DISCUSSION:  • The term 'statistically significant' should be used here (p.11, line 47), since it is not clear whether the reduction in sugar content was clinically significant. • There is no mention of the difference between own brand and branded products, and the fact that the sugar content of own brand products decreased more than that of branded products. This seems to highlight an important area for future intervention. • The paragraph on caffeine (p.12) implies that the only risk associated with caffeine is that it can increase SSB consumption. There are various risks associated with caffeine intoxication, addiction and withdrawal, particularly for children, which should be noted here. There should also be some mention of the aforementioned EU legislation relating to labelling of energy drinks. • The included text on measures taken to reduce sales of energy drinks has largely been taken from an FRC briefing paper (Visram and Hashem 2016) and therefore needs to be reworded or cited as direct quotations. I would advise removing much of this information and summarising the main messages into a sub-section on either 'comparison with the existing literature' or 'policy and practice implications'. • The paragraph from line 29, p.14, is lacking a citation for The Grocer article. • Any differences observed between supermarkets, convenience stores and health food stores should be discussed in the sub-section on limitations of the study. Are the authors confident that the same process was followed in 2015 and 2017? I am very interested to know if there are any other possible explanations for the reduction in the number of products surveyed between the two years. CONCLUSION  • The main message of the article could be clearer, in terms of how policy makers and practitioners should use the results. Presumably the message is about reformulation, particularly targeting well-known brands (while acknowledging the progress made so far in relation to supermarket own brand products)?
--	---

VERSION 1 – AUTHOR RESPONSE

Reviewer 1

Comment: This study seems to have been conducted with a particular objective in mind: to use data on the sugar content of energy drinks in UK grocery stores as a means to argue for product reformulation policies. The title therefore is misleading as to the content of the paper as calories and caffeine are not adequately discussed. The only reason for a survey so quickly (2017) after the previous one (2015) was to check for reformulation so maybe a paper with a clear focus on reformulation could be developed. Also the fact that there are 31 less drinks available after only 2 years is not referred to at all.

Response: Yes, the primary objective was to focus on reformulation of sugar and energy of energy drinks, but we are also concerned with the levels of caffeine in energy drinks, so we tried to capture this data more comprehensively in 2017. Since data on caffeine content of energy drinks has not been extensively reported in the literature, we feel it is important to include this data as it will be a

missed opportunity to not include this. We have changed the order of the introduction to focus on sugar first then on caffeine.

We added the following to the introduction section: Besides the health concerns around sugar content, some (but not all) 'energy' drinks contain high levels of caffeine, which is associated with chronic sleep loss, addiction/dependence, withdrawal and intoxication. [26, 30-32]

In terms of caffeine content, the EU Food Information Regulation requires specific labelling for high caffeine drinks (over 150mg per litre (mg/l)). [37] The warning states that the product is not recommended for children. However, these products can be easily purchased and consumed by children.

Comment: An even more pressing point and the reason why I chose to reject the article is that the term "energy drinks" has been used inappropriately – while there are many definitions as to exactly an energy drink encompasses, one similarity in all definitions is that they contain caffeine – something which the authors themselves state in line one of their introduction however in the methodology it is stated that any sugar sweetened drink which used the term energy, energise, stimulate or similar on its packaging was included. This is essentially then an additional review of a subset of soft drinks but not actually energy drinks. Any additional work on this needs to consider what to define this group as.

Response: Yes, we agree with the reviewer on this point. However, the energy drinks market is quite diverse and we wanted to capture all the products that are described as 'energy drinks' some of which may not contain high levels of caffeine but will still be seen by the industry and consumers as 'energy drinks'. The products containing high levels of caffeine, are only a sub-set of the energy drinks market, there are many other products, like Lucozade Energy range, which do not contain high caffeine warnings but are sold and purchased as 'energy drinks'. To be comprehensive it was important to include all the products.

We changed the definition in the introduction slightly and have clarified further the products included in our study in the methodology.

Introduction: Products described as 'Energy drinks' are typically non-alcoholic drinks containing a combination of ingredients such as caffeine and/or sugar and/or other ingredients such as guarana, taurine and ginseng, which are known to have stimulant properties and are distinct from sports drinks. [22, 26]

Methods:

Definition

'Energy' drinks were defined as any drink products with 'energy', 'energise', 'energiser', 'caffeine' and 'stimulation' in the product name or description e.g. Red Bull Energy Drink, Monster Energy Drink, Relentless Origin Energy Drink. Additionally, products that are known to contain high caffeine warnings but are not described as 'energy drinks' (such as Mountain Dew) were also included.

Since the focus of this study is on product reformulation, products labelled called 'zero' or 'light' or 'no added sugar' etc. were excluded. We also excluded products described solely as 'sports' drinks, which consist primarily of carbohydrates and electrolytes and are intended for athletes to rehydrate after exercise e.g. 'Lucozade Sports'. [22] However, 'Lucozade energy' was included.

Some additional concerns, in order of the layout of the manuscript:

1. Abstract – It is not clear what the numbers in the bracket refer to in the study design, I assume it is products surveyed and therefore this would be more appropriate in results.

Response: We deleted the numbers in the brackets.

Comment: Reference to the maximum daily recommendation for sugar is made in the results but then adults are used in the conclusions – be consistent.

Response: We removed this.

Comment: In addition, as with the closing conclusion of the paper it is a very broad and unsupported claim that “reducing sugar content of the drinks would be large in reducing both sugar and calorie consumption and therefore a reduction in obesity, type 2 diabetes and dental caries”

Response: The concluding statement was in regards to high consumers of soft drinks, including energy drinks but we have changed it to the following: In order to reduce the harmful impact of energy drinks, further reduction in sugar and a reduction in caffeine by reformulation are urgently needed. Other measures such as ban on the sale of energy drinks to children and smaller product sizes should also be explored, while warning labels should be kept. A reduction in sugar, energy and caffeine content and overall energy drinks consumption could be beneficial in reducing sugar, energy and caffeine intake of consumers of energy drinks.

2. There is a lack of referencing throughout the paper, I have included the lines here:

(page 3 lines 23, 24; page 4 lines 15, 18, 35; page 5 line 12; page 6 line 3, page 7 line 16; page 8 line 44, page 9 line 34; page 12 line 5, 16, 19, 56; page 13 line 17, 25, 31, 44, 52, 54)

Response: We updated the introduction and discussion sections and added the references where relevant.

3. Introduction – Focusses solely on sugar content therefore not sure why calories and caffeine are included in title – seems like an afterthought. Using the term free sugars may be confusing to a lay audience, after defining free sugars I think it would be okay to state that you will refer to these as sugar throughout. There is mix of tenses in some sentences e.g. lines 20-21.

Response: Please refer to previous comments. We stuck to using ‘sugar’ throughout manuscript and changed the tenses.

4. Method – the definition of energy drinks is not consistent with that of your first line in the introduction or of universally understood definitions. For serving size even when stated on a 500ml bottle label that 250ml was the serving size it was decided to include 500ml as a serving size however when dealing with the 1 litre bottles a 250ml serving size were used, aren’t people likely to also overconsume with this size bottle?

Response: Regarding definition, please refer to previous explanation.

We agree with this very good point, we removed the 1 litre bottle data from the per serving analysis, since we agree people tend to overconsume and won’t stick to the recommended 250ml serving size. We reanalysed the per serving data.

5. Results – Including the same brand in different serving sizes will not give an accurate picture of reformulation as one brand with 4 different serving sizes will give the appearance of several drinks changing their content.

Response: The same brand in different serving sizes is only included in the per serving data. We also did not report on the changes in the average per serving in the trend analysis (last section of the results section).

Comment: I think in future using per 100ml while stating the serving sizes available may be more comprehensive and easier to follow. There are five pages of results which is far too much and it is very repetitive and seems unclear in what it really wants to show. You have also included some material which may be more suited to the discussion in this section. On page 7 line 43 sugar should read calorie.

Response: Yes we use per 100ml data as the primary data and within this data only 1 formulation, regardless of serving size, is used. We have summarised the results pages to 3 pages.

Comment: When comparing the average sugar content to the maximum daily recommendation for sugar intake (page 10 line 49 onwards), you mention among the 65 products – it is not clear which 65 you are referring to as you have presented so many products. Also you need to include the word sugar in the brackets for line 52.

Including a list of the products as an appendix may have been useful

Response: We summarised the results, included more data in the tables for easy of reference and also included all the products in an appendix.

6. Discussion – this is started as energy drinks consistently surveyed, I'm not sure that twice constitutes consistently.

Response: We changed 'consistently' to 'products surveyed in both 2015 and 2017'.

Comment: You also state a significant reduction in sugar content however, my understand is this really refers to one brand with several different serving sizes?

Response: The significant reduction is based on per 100ml data, where only one formulation of each product is included. As mentioned in the results section, this is likely due to the 12 products reduced in sugar per 100ml. Please refer to last section of results section.

Comment: In line 18 page 14 it is mentioned that even small reductions in sugar content of energy drinks would have a significant impact on sugar and calorie intake of the population but this is not substantiated with evidence.

Response: Small reductions in high selling brands of energy drinks can have a big impact on sugar and energy intake of energy drink consumers. Ma et al, 2016, modelled the impact of reformulating soft drinks (includes energy drinks), which we reference.

Ma Y, He FJ, Yin Y, et al. Gradual reduction of sugar in soft drinks without substitution as a strategy to reduce overweight, obesity, and type 2 diabetes: a modelling study. *The Lancet Diabetes & Endocrinology*. 2016;4:105-14.

7. Conclusion – There is information here not previously mentioned anywhere in the paper such as pervasive and misleading marketing targeted at young people.

"This study indicates there has been a reduction in sugar content of energy drinks in the UK" – there has only been a reduction in some or most likely one main brand. In addition, regarding the last line of the conclusion it is very naïve statement to assume that product reformulation would lead to a very large reduction in sugar and calorie consumption and therefore a reduction in obesity, type 2 diabetes and dental caries. Obesity in particular is a multi-faceted and complex condition and it will take much

more than reformulation of energy drinks to make an impact on levels. Also it is not clear that this would change population dietary patterns as substitution is likely.

Response: The concluding statement was in regards to high consumers of soft drinks, including energy drinks but we have changed it to the following: In order to reduce the harmful impact of energy drinks, further reduction in sugar and a reduction in caffeine by reformulation are urgently needed. Other measures such as ban on the sale of energy drinks to children and smaller product sizes should also be explored, while warning labels should be kept. A reduction in sugar, energy and caffeine content and overall energy drinks consumption could be beneficial in reducing sugar, energy and caffeine intake of consumers of energy drinks.

Comment: I think that the authors may be able to reuse some of the data to look solely at reformulation and to be more clear about that at the beginning, however in its current format I don't think this paper is suitable for publication.

Response: We have modified the paper based the reviewers comments and feel it has improved significantly now.

Reviewer: 2

Reviewer Name: Shelina Visram

Institution and Country: Associate professor in public policy and health, Durham University, UK

Please state any competing interests or state 'None declared': None declared

Please leave your comments for the authors below

Thank you for the opportunity to review this manuscript, which I found interesting and relevant to my own work. There has been little research to date on the contents of commercial energy drinks, which continue to increase in popularity particularly amongst young consumers. The authors' analyses appear to be robust and the results are reported in some detail. However, there are a number of areas where I believe the manuscript could be strengthened in order to make it suitable for publication.

Thank you for reviewing our manuscript.

INTRODUCTION:

- The text in this section could be reordered to tell a clearer story, beginning with the SACN recommendation, a clear definition of free sugars, the role of soft drinks and then drilling down to energy drinks as a specific sub-category of soft drink.

Response: We reordered the text to focus on sugar.

- The included text on energy drinks has largely been taken from either a previous BMJ Open paper (Visram et al 2016) or an FRC briefing paper (Visram and Hashem 2016) and therefore needs to be replaced, reworded or cited as direct quotations.

Response: We updated and reworded the introduction section.

- The authors refer to 'Another study by EFSA' (p. 3, line 40) but this is actually the same study as that cited in the previous sentence. The paper by the Nomisa-Areté Consortium is the original source for this study.

Response: We corrected this.

- The terms 'free sugars' and 'sugars' are used throughout the manuscript. While this may be technically correct, 'sugar' (singular) is more appropriate. For example, it should be 'sugar-sweetened beverages', not 'sugars-sweetened beverages'.

Response: We changed the text to use 'sugar' alone.

- This section makes a convincing case for investigating the sugar and calorie content of energy drinks, in light of the SACN recommendation and planned sugar levy. However, no rationale is given for examining data in relation to caffeine.

Response: We changed the introduction to reflect why our study primarily focuses on sugar and then caffeine.

METHODS:

- The data collection process is unclear – how exactly were data collected, e.g. in store or online? Using a predefined template? This section states that 'All data was double checked after entry' (p.5, line 29) – entry into what? Checked by whom?

Response: The data was collected in store. The predefined 'template' is stated in the method already here: For each 'energy' drink, the data collected included the company name, product name, pack weight, serving size, sugars (g), energy (kcal) and caffeine (mg) per 100ml and sugars, energy and caffeine per serving. Where data was not available per serving, it was calculated from pack size and per 100ml data.

The data was entered on an excel spreadsheet and double checked by the first author.

- What was the rationale for including 'beauty and health food retailers' in this study? I fear this may have skewed some of your results (see below).

A pilot survey of energy drinks showed that many energy drinks products are sold in health and beauty shops too, so we included them, therefore the survey is as comprehensive as possible.

Response: We are confident this wouldn't have skewed the data, but in case it has, we added a sub-section to the results table looking at only energy drinks with high caffeine warning label, which are traditionally sold in the supermarkets. There isn't a major difference between the two data-sets.

- 'Convenience store' needs to be added in brackets after 'a Costcutter store' (p.5, line 42).

Response: We added this.

- Has the term 'supermarket own brand' been used to refer to 'own brand' products on sale in convenience and health food stores as well as supermarkets? If so, this is confusing and needs rewording. If not, what was the rationale for only including supermarket products?

Response: We changed this to 'own label'. Health food stores and convenience stores do not sell own-label energy drinks, only branded.

- Related to a previous point, it should be stated explicitly somewhere in the manuscript that sugar was the primary concern in this study, with caffeine a secondary concern. This would then help to

clarify the rationale for excluding 'no added sugar' drinks, which are still likely to be high in caffeine and in need of reformulation.

Response: We updated the title to the following and reordered the introduction too.

Title: Cross-sectional surveys of the amount of sugar, energy and caffeine in sugary energy drinks in the UK between 2015 and 2017

Response: We have also added the following: Since the focus of this study is on product reformulation, products labelled called 'zero' or 'light' or 'no added sugar' etc. were excluded. We also excluded products described solely as 'sports' drinks, which consist primarily of carbohydrates and electrolytes and are intended for athletes to rehydrate after exercise e.g. 'Lucozade Sports'. [22] However, 'Lucozade energy' was included.

- On reading your definition of an energy drink, readers may be confused if they are familiar with products such as Red Bull or Monster which do not seem to have 'energy' in the name. Please include one or two examples to reduce confusion, e.g. 'Monster energy drink'.
- You state that you excluded isotonic or sports drinks but presumably this was not true if they met the inclusion criteria, e.g. did you include Lucozade Energy and, if not, why not?

Response: We have changed the definition in the introduction slightly and have clarified further the products included in our study in the methodology. Please refer to previous comments.

- The sub-sections on 'Serving size' and 'Per 100ml verses per serving' could be combined (also 'verses' should be changed to 'versus').

Response: We clarified this to the following: Per 100ml: Some brands sell the same formulation in different serving sizes. Therefore, the 100ml data only included an example of one formulation regardless of the different serving sizes.

Per serving: The per serving data included all the different serving sizes available apart from 1 litre bottles. Any product with up to 500ml can or bottle size was considered as one serving, regardless of what is stated on the packaging as a serving size e.g. often a 500ml bottle is split into two servings but we consider that most consumers drink these drinks as one serving. 1 litre products were excluded from the per serving analysis since it was deemed that the industry standardised serving of 250ml was too little and that consumers are likely to overconsume.

- The tense used in the sub-section on 'High, medium and low criteria' should be changed from present/future to past tense.

Response: We corrected this.

RESULTS:

- Throughout this section (and the manuscript as a whole), the information on caffeine reads as an afterthought. It would be preferable to either leave it out altogether or do more to highlight its importance. There is no mention of caffeine in the tables or figures.

Response: We added caffeine to the tables and conducted further analysis on the products with high caffeine warning label.

- It is not clear why the numbers of products surveyed are different in the sub-sections on 'per 100ml' and 'per serving'. Did some of the product labels lack information on sugar and calories per 100ml? If

so, it would be useful to identify the types of products affected (given that they should all presumably be subject to the same food labelling regulations).

Response: Please refer to reply in on per 100ml vs per serving in the methods section above.

- Similarly, it would be helpful to know more about products that were not labelled with their caffeine content or the 'high caffeine' warning. Energy drink labels are required to include this information by EU legislation in place since 2011.

Response: These products generally contain no caffeine (but described as 'energy drinks') or the caffeine does not exceed the warning label cut-off, explained here in the introduction, method and discussion section:

Introduction: In terms of caffeine content, the EU Food Information Regulation requires specific labelling for high caffeine drinks (over 150mg per litre (mg/l)). [37] The warning states that the product is not recommended for children. However, these products can be easily purchased and consumed by children.

Method: Caffeine: Separate analysis was conducted on the products with the high caffeine warning label (over 150mg per litre (mg/l)).

Discussion: It important to note that not all manufacturers label the amount of caffeine in their energy drinks and this is likely because their products do not exceed the cut-off for adding the warning label.[37] Among the companies that do, a warning is stated on the pack, acknowledging the potential danger these products can pose to children. However, these products can be easily purchased and consumed by children.

- The low values given here seem very low and I wonder if these reflect the inclusion of products sold in health food stores, which seem like a different category of products that are less likely to be used by children and adolescents. Were they sold primarily as food or exercise supplements?

Response: Please refer to responses earlier and products list in appendix. They are not food or exercise supplements.

- It would be good to know more about the products with very low and very high levels of sugar, as well as those that were reformulated between the two time periods. The tables only provide averages for own brand and branded products, which are useful, but it would be good if they could also include information on the top and bottom point of each range.

Response: We added more data such as in table 2 and table 3.

- This section is a little hard to read and I wonder if it would be easier to follow if sub-sections were combined across the two years, i.e. sugar / calories / caffeine / per 100ml in 2015 vs 2017 and per serving size in 2015 vs 2017. There is a separate sub-section comparing the same products between the two years but it is also interesting to note overall changes across all products in this period.

Response: We summarised the results section. The overall changes in all products is shown in the earlier sections and in table 1.

DISCUSSION:

- The term 'statistically significant' should be used here (p.11, line 47), since it is not clear whether the reduction in sugar content was clinically significant.

Response: 'Statistically' has been added.

- There is no mention of the difference between own brand and branded products, and the fact that the sugar content of own brand products decreased more than that of branded products. This seems to highlight an important area for future intervention.

Response: The decrease appears to be more in the supermarket products vs branded but this was not statistically significant, so we did not report it. We can report it, if the reviewers feel it is necessary.

- The paragraph on caffeine (p.12) implies that the only risk associated with caffeine is that it can increase SSB consumption. There are various risks associated with caffeine intoxication, addiction and withdrawal, particularly for children, which should be noted here. There should also be some mention of the aforementioned EU legislation relating to labelling of energy drinks.

Response: We updated the introduction with the following: Besides the health concerns around sugar content, the high levels of caffeine in some energy drinks are associated with chronic sleep loss, addiction/dependence, withdrawal and intoxication. [26, 30-32]

And the discussion with the following: There is also some evidence to suggest that the caffeine content increases sugar-sweetened drinks consumption further and therefore sugar intake too.[46] Since children and teenagers are high consumers of energy drinks, manufacturers should consider reducing levels of caffeine - again through reformulation. Indeed a recent study showed that removing caffeine from sugar-sweetened drinks, along with 10.3% of sugar, has the potential to reduce body weight of adults by 0.6kg, without any change in sugar-sweetened drinks consumption. [47]

- The included text on measures taken to reduce sales of energy drinks has largely been taken from an FRC briefing paper (Visram and Hashem 2016) and therefore needs to be reworded or cited as direct quotations. I would advise removing much of this information and summarising the main messages into a sub-section on either 'comparison with the existing literature' or 'policy and practice implications'.

Response: This has been summarised to the following: Perhaps the UK should consider restrictions like other countries have. In Lithuania, sales of energy drinks (containing at least 150mg of caffeine per litre) are banned to under-18s.[48] A similar law applies in Sweden and sales of some types of energy drinks are restricted too.[26]

- The paragraph from line 29, p.14, is lacking a citation for The Grocer article.

Response: This has been added.

- Any differences observed between supermarkets, convenience stores and health food stores should be discussed in the sub-section on limitations of the study.

Response: We can't discuss this, since when the data was collected we did not identify the products against store type, only whether it was own label or branded.

Comment: Are the authors confident that the same process was followed in 2015 and 2017? I am very interested to know if there are any other possible explanations for the reduction in the number of products surveyed between the two years.

Response: The same process was followed, apart for caffeine data collected in 2015 (since at the time caffeine was not the primary purpose of the survey), which we explain in the methods section: For each 'energy' drink the data collected included the company name, product name, pack weight, serving size, sugars (g), energy (kcal) and caffeine (mg) per 100ml and sugars, energy and caffeine per serving. Where data was not available per serving, it was calculated from pack size and per 100ml data. Caffeine content was only collected in a sub-sample of the 2015 products.

Also, we have added the following to the discussion section to explain the possible reduction in the number of products.

The surveys have shown early changes in the energy drinks market. Formulations (per 100ml) and number of products (per serving) have fallen between 2015 and 2017. This may be due to the pressure on the soft drinks industry to reduce sugary drinks offerings. [42] Furthermore, the energy drinks surveyed showed a small but statistically significant reduction in sugar content; this is likely due to the 15 products which have reduced sugar content between 2015 and 2017, probably as a result of reformulation due to the SDIL. [35]

CONCLUSION

- The main message of the article could be clearer, in terms of how policy makers and practitioners should use the results. Presumably the message is about reformulation, particularly targeting well-known brands (while acknowledging the progress made so far in relation to supermarket own brand products)?

Response: We updated the conclusion to the following: Factors such as an increase in sales, [24, 25, 27] concerning levels of sugar, energy and caffeine (assessed in this study), ease of access [33] as well as limited regulation have created an environment where energy drinks could pose a major threat to public health.

In order to reduce the impact of energy drinks, sugar (and therefore energy) reformulation should continue and begin regarding caffeine content. Other measures such as a ban on the sale of energy drinks to children should be explored, while warning labels should be kept. Can and bottle sizes of energy drinks should also be reduced to reduce overall consumption of sugar, energy and caffeine. A reduction in sugar, energy and caffeine content and overall energy drinks consumption could be beneficial in reducing sugar, energy and caffeine intake of consumers of energy drinks

VERSION 2 – REVIEW

REVIEWER	Laura Keaver Institute of Technology Sligo Ireland
REVIEW RETURNED	26-Aug-2017

GENERAL COMMENTS	Thank you for giving me the opportunity to re-review this article which has been much improved and given a more focussed aim since the initial submission. I have included my comments below: Sugary energy drinks is not a definition and may not be suitable for publication. Page 2 Line 15: remove 'sugars' from the brackets Page 3 Line 32: Average intake is mentioned however it is not clearly where this is or when it was recorded. Also add 'daily' to energy intake.
---

	Page 3 Line 38: do the new recommendations refer to SACN? Maybe include a section on reformulation in the introduction Page 5 Line 13-15: Label the last aim as (e) - be consistent, also these should match up then directly with how the results are presented. Page 5 Line 29: Only half a bracket used Page 5 Definition: It is important to be clear here that several products did not include caffeine which is usually used as a definition. Maybe state that caffeine OR the use of words such as energy, stimulant etc. were used and therefore the majority of products actually did not contain caffeine. Page 5 Line 47: remove 'called' Page 6 Line 5: inverted commas around energy which isn't done elsewhere Page 6 Line 19: sugar content was assumed the same as in 2017 if unavailable for 2015 - how many products was this done for? Page 7 Line 28: only provide % daily energy intake for highest drink, consider giving range Page 8 Line 25: using energy and kcal - maybe use just one Page 8 Line 44: it is mentioned that not all products were collected in 2015 - please elaborate on this and its potential impact The heading 'per 100ml' under results could be retitled as 'reformulation'. Try to match up the result headings with the aims for greater clarity. Page 11 Line 19: typo with 'manufacturers' Page 11 Line 37: reword 'with 95% would receive a' - remove 'with' Page 12 Line 5: This effect is likely due to the reduction in sugar and subsequently energy intake rather than the caffeine content
--	--

REVIEWER	Shelina Visram Newcastle University, UK The lead author and I co-authored a briefing paper entitled "Energy drinks: what's the evidence?", which was published by the Food Research Collaboration in July 2016.
REVIEW RETURNED	29-Aug-2017

GENERAL COMMENTS	Thank you for revising this manuscript and taking on board many of the reviewers' comments, which I feel have helped to strengthen the paper. There are, however, some comments which have not been fully addressed, as well as a small number of new issues relating to the revised text:  • The main issue relates to the way in which the term 'energy drink' has been defined in this study. As highlighted by Reviewer One previously, the defining characteristic of what most people understand by the term 'energy drink' is that it contains caffeine, and in high doses. However, this study includes a number of drinks that
---

	contain little or no caffeine. The definition provided in the introduction (p.3, lines 49-54) applies to all soft drinks (i.e. those containing caffeine or sugar or other ingredients), but states that these products are distinct from sports drinks (without specifying in what way). Yet the definition and examples given in the methods section include sports drinks and also enhanced fruit juice. In your response to the previous comments, you state that you set out to include all products described or seen by the industry and consumers as 'energy drinks', and yet you included products such as Mountain Dew that are not described in this way (p.5, lines 41-43). As a minimum, I would suggest using the term 'beverages marketed or consumed as energy drinks' in your title (rather than 'sugary energy drinks'), and also discussing this as a limitation of your study (in terms of having treated high and low caffeine drinks as one category of soft drinks). A clear definition should be introduced early on and used consistently throughout, rather than having two or three conflicting definitions (given that the definition used in UK legislation – products containing at least 150mg of caffeine per litre – is also included on p. 12, line 23).  • In my previous review I queried the following statement from the methods section: 'All data was double checked after entry' (p.6, lines 23-24) – entry into what? Checked by whom? If the data were* entered into an Excel spreadsheet, it would be helpful to state this explicitly, otherwise the sentence looks a bit odd (*also, 'data' should be plural rather than singular; this is incorrect throughout the manuscript). • The section on limitations of the study could be strengthened, rather than highlighting a single limitation (i.e. relying on the accuracy of data provided on drink labels). For example, collecting information on caffeine from some products but not others, and changing the data collection process slightly between the two time periods could be discussed here. • The table given in the appendix is very helpful, but it is not clear whether the blank spaces relate to details that were not collected (e.g. data on caffeine not collected in 2015) or information that was not present on the label. It is surprising to see so many products branded as 'energy drinks' that do not provide information on their caffeine content. • The abstract does not mention convenience stores.
--	--

VERSION 2 – AUTHOR RESPONSE

Reviewer 1

Comment: Sugary energy drinks is not a definition and may not be suitable for publication.

We feel this paper is suitable for publication since it makes a significant contribution to the limited research on energy drinks and also exposes some misconceptions such as the definitions and what is available to consumers.

Response: 'Sugary energy drinks' is not a definition; it is just a short form for our detailed definition. We are clarifying the three definitions here:

Industry definition - Traditional glucose based energy drinks; functional or stimulation energy drinks which claim a particular energy boost from caffeine, guarana, taurine, ginseng or other herbs or some combination of these ingredients.

EU legislation definition – high caffeine drinks (over 150mg per litre (mg/l)).

Our definition – Energy drinks were defined as any drink products with ‘energy’, ‘energise’, ‘energiser’, ‘caffeine’ and ‘stimulation’ in the product name or description e.g. Red Bull Energy Drink, Monster Energy Drink, Relentless Origin Energy Drink and Tropicana Energy Mango and Guava with Passion fruit OR products with high caffeine warning label including products not described as energy drink (such as Mountain Dew). Our definition takes into account both definitions.

We also dispute the main assumption made by the reviewers that the defining characteristic of what most people understand by the term ‘energy drink’ is that it contains caffeine, and in high doses. The definition you are referring to describes only a subgroup of energy drinks products. For instance I am sure you consider Lucozade Energy drinks range as an energy drinks, and the majority of consumers do too, and the industry does too but this big market leading product range does not contain high levels of caffeine and possibly has never had to include a high caffeine label, because the level of caffeine does not exceed 150mg per litre. Therefore, according to the reviewer’s narrow definition of energy drinks, Lucozade Energy product range will be excluded from our analysis. This is not our intention. We intend to include a broader definition of energy drinks, as stated by the soft drinks industry and further highlight, as we do, those products with high caffeine labels.

Furthermore, we are concerned with the sugar content of products containing sugar and are high in caffeine and/or described as ‘energy drinks’ because they can be reformulated.

Nevertheless, to further clarify the scope of our survey, we have made the following changes:

1. We have changed the title to ‘sugary soft drinks marketed and consumed as energy drinks’.
2. We have also included the industry definition in the introduction as our leading definition and later stated that some of these products need to highlight on pack the level of caffeine, according to the EU legislation, we have therefore included both.

Page 2 Line 15: remove 'sugars' from the brackets We think this should be kept, since ‘sugars’ not ‘sugar’ is what is included on nutrition panel of products.

Page 3 Line 32: Average intake is mentioned however it is not clearly where this is or when it was recorded. We have added the year.

Also add 'daily' to energy intake. We added this.

Page 3 Line 38: do the new recommendations refer to SACN? We removed this, since the first sentence was stating something similar.

Maybe include a section on reformulation in the introduction We added this.

Page 5 Line 13-15: Label the last aim as (e) - be consistent, also these should match up then directly with how the results are presented. We added (e).

Page 5 Line 29: Only half a bracket used Removed.

Page 5 Definition: It is important to be clear here that several products did not include caffeine which is usually used as a definition. Maybe state that caffeine OR the use of words such as energy, stimulant etc. were used and therefore the majority of products actually did not contain caffeine. Definition updated to add ‘OR’. Also, in 2017 the majority of the products did contain caffeine (42/59).

Page 5 Line 47: remove 'called' Removed.

Page 6 Line 5: inverted commas around energy which isn't done elsewhere Removed.

Page 6 Line 19: sugar content was assumed the same as in 2017 if unavailable for 2015 - how many products was this done for? Only one.

Page 7 Line 28: only provide % daily energy intake for highest drink, consider giving range Since our focus is on reformulation, we think it is sufficient to illustrate only the highest products and imply that they need to be brought down. Ranges of energy content are provided in the tables and illustrate lower levels are available.

Page 8 Line 25: using energy and kcal - maybe use just one Replaced with energy.

Page 8 Line 44: it is mentioned that not all products were collected in 2015 - please elaborate on this and its potential impact

We've already explained in the method section that - Caffeine content was only collected in a sub-sample of the 2015 products – and added further explanation in the limitation section.

The heading 'per 100ml' under results could be retitled as 'reformulation'. Try to match up the result headings with the aims for greater clarity.

Retitled and reordered the aims.

Page 11 Line 19: typo with 'manufacturers' Corrected.

Page 11 Line 37: reword 'with 95% would receive a' - remove 'with' Removed.

Page 12 Line 5: This effect is likely due to the reduction in sugar and subsequently energy intake rather than the caffeine content Corrected.

Reviewer: 2

Reviewer Name: Shelina Visram

Please leave your comments for the authors below Thank you for revising this manuscript and taking on board many of the reviewers' comments, which I feel have helped to strengthen the paper. There are, however, some comments which have not been fully addressed, as well as a small number of new issues relating to the revised text:

Comment: The main issue relates to the way in which the term 'energy drink' has been defined in this study. As highlighted by Reviewer One previously, the defining characteristic of what most people understand by the term 'energy drink' is that it contains caffeine, and in high doses.

Please refer to earlier comments.

However, this study includes a number of drinks that contain little or no caffeine. The definition provided in the introduction (p.3, lines 49-54) applies to all soft drinks (i.e. those containing caffeine or sugar or other ingredients), but states that these products are distinct from sports drinks (without

specifying in what way). Please refer to earlier comments. We have specified the differences between energy and sports products now.

Yet the definition and examples given in the methods section include sports drinks and also enhanced fruit juice. No they do not include any 'sports' drinks. We would be grateful if you can point out the 'sports' drinks. Also, juices with 'energy' in the product name, likely because they have added guarana, are within our definition, so they must be included.

In your response to the previous comments, you state that you set out to include all products described or seen by the industry and consumers as 'energy drinks', and yet you included products such as Mountain Dew that are not described in this way (p.5, lines 41-43). Yes but we also stated that we included products with high caffeine warning label and Mountain Dew contained the high label in 2015, so accounting to our definition it should be included.

As a minimum, I would suggest using the term 'beverages marketed or consumed as energy drinks' in your title (rather than 'sugary energy drinks'), and also discussing this as a limitation of your study (in terms of having treated high and low caffeine drinks as one category of soft drinks). A clear definition should be introduced early on and used consistently throughout, rather than having two or three conflicting definitions (given that the definition used in UK legislation – products containing at least 150mg of caffeine per litre – is also included on p. 12, line 23).

Response: We have not included conflicting definitions. Please refer to previous comments. Also, the reviewer's previous comments suggested we highlight that we included only sugary versions of energy drinks, hence why we added the word 'sugary', we feel this is important, but now you suggest we remove it. Nevertheless, we have updated and clarified the definition further.

Comment: In my previous review I queried the following statement from the methods section: 'All data was double checked after entry' (p.6, lines 23-24) – entry into what? Checked by whom? If the data were* entered into an Excel spreadsheet, it would be helpful to state this explicitly, otherwise the sentence looks a bit odd (*also, 'data' should be plural rather than singular; this is incorrect throughout the manuscript).

It was checked by the lead author, and this is stated in the contributions. It is obvious that the data would be double checked. We have updated the text to the following - The data were double checked after entry into excel spreadsheet, and a further 5% of entries were checked against the original source in a random selection of products by the lead author.

Comment: The section on limitations of the study could be strengthened, rather than highlighting a single limitation (i.e. relying on the accuracy of data provided on drink labels). For example, collecting information on caffeine from some products but not others, and changing the data collection process slightly between the two time periods could be discussed here.

Response: We strengthen the limitation section to highlight the lack of caffeine data in 2015, as suggested.

Comment: The table given in the appendix is very helpful, but it is not clear whether the blank spaces relate to details that were not collected (e.g. data on caffeine not collected in 2015) or information that was not present on the label. It is surprising to see so many products branded as 'energy drinks' that do not provide information on their caffeine content.

Response: We have added 'NC' for not collected, 'NA' for products that did not exist in 2015, so caffeine data collection was not applicable and 'NL' for no caffeine warning label.

Comment: The abstract does not mention convenience stores. We added this.

VERSION 3 – REVIEW

REVIEWER	Laura Keaver Institute of technology Sligo
REVIEW RETURNED	07-Sep-2017

GENERAL COMMENTS	My comments below: Page 2 line 52: maybe change to 'marketed as energy drinks' Page 3 line 32: add g to 54 Page 7 line 42: put in past tense Page 11 line 38: impending SDIL Page 11 line 47: have you any figures on high consumers i.e. % of the population who are high consumers Page 11 line 50: early success even though not implemented yet? Might be more accurate to discuss how though not implemented that awareness of it has led to early reformulation to avoid tax when introduced Page 12 line 26: without affecting taste rather than with affecting taste Page 12 line 40: In Lithuania, sales of energy drinks (containing at least 150mg of caffeine per litre) are banned to under-18s. [49] A similar law applies in Sweden and sales of some types of energy drinks are restricted too.[26] - second line makes no sense you are discussing restrictions of energy drinks then say some types of energy drinks are restricted too Page 13 line 35: collected rather than collect Page 13 line 32: what was the initial aim and will this difference affect and skew the new aim and subsequent results?
--

VERSION 3 – AUTHOR RESPONSE

Reviewer: 1

Reviewer Name: Laura Keaver

Institution and Country: Institute of technology Sligo Please state any competing interests or state

'None declared': None declared

Please leave your comments for the authors below My comments below:

Page 2 line 52: maybe change to 'marketed as energy drinks'

Amended.

Comment: Page 3 line 32: add g to 54

Response: Added

Comment: Page 7 line 42: put in past tense

Response: Amended.

Comment: Page 11 line 38: impending SDIL

Response: Amended.

Comment: Page 11 line 47: have you any figures on high consumers i.e. % of the population who are high consumers

Response: We don't have access to this type of data.

Comment: Page 11 line 50: early success even though not implemented yet? Might be more accurate to discuss how though not implemented that awareness of it has led to early reformulation to avoid tax when introduced

Response: Amended.

Comment: Page 12 line 26: without affecting taste rather than with affecting taste

Response: Amended.

Comment: Page 12 line 40: In Lithuania, sales of energy drinks (containing at least 150mg of caffeine per litre) are banned to under-18s. [49] A similar law applies in Sweden and sales of some types of energy drinks are restricted too.[26] - second line makes no sense you are discussing restrictions of energy drinks then say some types of energy drinks are restricted too

Response: This should say 'the sale of energy drinks are restricted to pharmacies' - we amended the whole sentence.

Comment: Page 13 line 35: collected rather than collect

Response: Amended.

Comment: Page 13 line 32: what was the initial aim and will this difference affect and skew the new aim and subsequent results?

Response: As described the initial aim (2015) focused on collecting sugar and energy content, yes we suggested that the caffeine data for 2015 may not be a full reflection of caffeine content of energy drinks on the market in 2015.